# Occurrence of Plasmid-Mediated Quinolone Resistance and Carbapenemase-Encoding Genes in *Pseudomonas aeruginosa* Isolates from Nosocomial Patients in Aguascalientes, Mexico

**DOI:** 10.3390/pathogens13110992

**Published:** 2024-11-13

**Authors:** Ana S. Tapia-Cornejo, Flor Y. Ramírez-Castillo, Alma L. Guerrero-Barrera, Diana E. Guillen-Padilla, José M. Arreola-Guerra, Mario González-Gámez, Francisco J. Avelar-González, Abraham Loera-Muro, Eduardo Hernández-Cuellar, Carmen L. Ramos-Medellín, Cesar Adame-Álvarez, Ricardo García-Romo, Fabiola Galindo-Guerrero, Adriana C. Moreno-Flores

**Affiliations:** 1Departamento de Medicina Interna, Hospital Centenario Miguel Hidalgo, Aguascalientes 20240, Mexico; srhitap.cor@hotmail.com; 2Laboratorio de Biología Celular y Tisular, Departamento de Morfología, Centro de Ciencias Básicas, Universidad Autónoma de Aguascalientes, Aguascalientes 20100, Mexico; flor.ramirez@edu.uaa.mx (F.Y.R.-C.); guillendiana2018@gmail.com (D.E.G.-P.); edgar.hernandez@edu.uaa.mx (E.H.-C.); fabiola.galindo@edu.uaa.mx (F.G.-G.); cecilia.morenof@edu.uaa.mx (A.C.M.-F.); 3Departamento de Nefrología, Hospital Centenario Miguel Hidalgo, Aguascalientes 20240, Mexico; dr.jmag@gmail.com; 4Departamento de Infectología, Hospital Centenario Miguel Hidalgo, Aguascalientes 20240, Mexico; mariogzg@hotmail.com; 5Laboratorio de Estudios Ambientales, Departamento de Fisiología y Farmacología, Centro de Ciencias Básicas, Universidad Autónoma de Aguascalientes, Aguascalientes 20100, Mexico; javier.avelar@edu.uaa.mx; 6CONAHCYT, Centro de Investigaciones Biológicas del Noreste (CIBNOR), La Paz 23205, Mexico; aloera@cibnor.mx; 7Laboratorio Clínico, Hospital Centenario Miguel Hidalgo, Aguascalientes 20240, Mexico; clramism23@gmail.com (C.L.R.-M.); adameces@hotmail.com (C.A.-Á.); ricardoromo547@gmail.com (R.G.-R.)

**Keywords:** carbapenemase-encoding genes, Plasmid-Mediated Quinolone Resistance (PMQR) genes, *mcr-1*, *Pseudomonas aeruginosa*, multidrug-resistance

## Abstract

*Pseudomonas aeruginosa* is a leading cause of healthcare-associated infections, which are related to substantial morbidity and mortality. The incidence of Plasmid-Mediated Quinolone Resistance (PMQR) determinants has been previously reported in this bacterium. However, there is limited information regarding the presence of PMQR and carbapenemase-encoding genes simultaneously. This study aims to analyze the prevalence of these determinants on *P. aeruginosa* strain isolated from clinical patients in the State of Aguascalientes, Mexico. Fifty-two *P. aeruginosa* isolates from nosocomial patients were collected from Centenario Hospital Miguel Hidalgo. This is a retrospective observational study conducted at a single center. Antibiotic susceptibility was tested using the Vitek-2 system. Only carbapenem-resistant isolates were included in this study. Carbapenemase-encoding genes and PMQR determinants were screened by polymerase chain reaction (PCR). Resistance rates of 100% were found on tigecycline and ceftriaxone. Of the 52 isolates, 34.6% were positive for the *qnr* genes, 46.2% for the *oqxA* gene, and 25% for the *aac-(6′)-lb* gene. The most frequent carbapenemase genes found in the samples were *bla*_OXA-51_ (42.3%), *bla*_OXA-1_ (15.4%), and *bla*_VIM_ (15.4%). *bla*_OXA-51_ co-carrying *oqxA* was detected in 21.1% of the isolates, *bla*_OXA-51_ co-carrying *aac-(6’)-lb* in 11.5%, *bla*_VIM_ co-carrying *aac-(6′)-lb* in 3.8%, and *bla*_KPC_ co-carrying *oqxA* in 5.8%. Systematic surveillance to detect carbapenemase-encoding genes and PMQR determinants, and rational prescription using the last-line drugs could help in preventing the dissemination of multidrug-resistant determinants.

## 1. Introduction

*Pseudomonas aeruginosa* is a Gram-negative bacterium that causes significant healthcare-related illnesses such as bacteremia, catheter-related infections, surgical site infections, ventilator-associated pneumonia, otitis, and urinary tract infections [1]. *P. aeruginosa* is naturally resistant to many common antibiotics due to its reduced expression of high-permeability porins and the acquisition of new resistance determinants. As a result of multi-drug resistance (MDR), infections due to these bacteria became challenging to treat [2], leading to infection outbreaks, longer hospital stays, and higher mortality rates [3].

Antimicrobial resistance (AMR) is one of the major health threats worldwide. It is estimated that bacterial AMR was directly responsible for 1.27 million global deaths in 2019 [4]. The development of MDR limits the treatment options since bacteria could display resistance to a broad spectrum of antibiotic classes, including aminoglycosides (amikacin, gentamicin, and tobramycin), fluoroquinolones (ciprofloxacin, ofloxacin, and norfloxacin), carbapenems, tetracyclines [5], and colistin [6]. Carbapenem antibiotics are the last resort for therapy for infections caused by MDR-*P. aeruginosa*; however, increased carbapenem-resistant *P. aeruginosa* (CRPA) has been reported [7,8]. Those mechanisms of carbapenem resistance include efflux of the drug mediated by overexpression of the MexAB-OprM efflux pump, overproduction of AmpC beta-lactamase, inactivation of the OprD outer membrane protein, and production of carbapenemases [9]. Class B carbapenemases or metallo-β-lactamases (MBLs) are major determinants of transferable resistance and include Verona integron-encoded MBLs (VIM) and imipenemase enzymes (IMP) [10]. In Mexico, *bla*_VIM_, *bla*_IMP_, and *bla*_GES_ are the most frequent carbapenemase-encoding genes found in *P. aeruginosa* [11,12,13].

Fluoroquinolones are widely used to treat infections caused by this bacterium. Quinolone resistance is primarily caused by a chromosomal mutation in the quinolone resistance-determining region (QRDR) of the DNA gyrase (*gyrA* and *gyrB*) and topoisomerase (*parC* and *parE*) encoding genes [14]. However, plasmid-mediated quinolone resistance genes (PMQR) have also been detected in *P. aeruginosa* [15,16,17,18,19,20,21,22,23,24]. PMQR-mediated mechanisms include the following: *qnr* genes, that encode the Qnr protein family (QnrA, QnrB, QnrS, QnrC, QnrD, QnrE, and QnrVC) [14]; the acetyltransferase *aac-(6′)-lb-cr,* which is a variant of an enzyme involved in aminoglycoside acetylation; and the active efflux pumps such as QepA and OqxA [14,21]. PMQR determinants can spread vertically and horizontally among bacteria and facilitate the selection of resistant mutants conferring only low-level fluoroquinolone resistance, which may not be detected by standard susceptibility testing but may increase the frequency of mutations in the chromosomal genes, facilitating the selection of high-level resistance to this antibiotic class [25].

Indeed, quinolone resistance determinant QnrVC has also been linked to some epidemic strains with acquired carbapenemases, such as ST175 and ST244 [26]. Furthermore, the presence of these genes might affect the minimal inhibitory concentration (MIC) to unrelated agents such as novobiocin, tigecycline, or colistin [27]. In addition, several authors have suggested that the presence of the *qnr* genes is associated with a trend related to extended hospital stays and increased 30-day mortality [27,28]. Thus, the co-occurrence of encoding carbapenemase-genes and PMQR determinants may facilitate the emergence of MDR isolates [29], amplifying antibiotic resistance and limiting the effectiveness of empirical therapy, resulting in the need for more potent and potentially toxic antibiotics such as polymyxins [30]. Additionally, novel agents like ceftazidime-avibactam, ceftolozane-tazobactam, imipenem-cilastatin-relebactam, and cefiderocol may be required [31], however, more clinical trials involving the news therapies are needed to answer these questions.

Given the limited data on the co-existence of PMQR and carbapenemase-encoding genes among clinical isolates of *P. aeruginosa* worldwide [22,23,24], and particularly in Mexico [32], the characterization of the carbapenemase-encoding genes and evaluation of PMQR determinants in *P. aeruginosa* isolates from nosocomial patients in Aguascalientes, Mexico, was carried out. Moreover, the colistin resistance *mcr-1* gene was also assessed.

## 2. Materials and Methods

### 2.1. Sampling and Bacterial Isolation

A total of 52 bacteria isolates were collected from July 2019 to February 2023 from nosocomial patients at Centenario Hospital Miguel Hidalgo, Aguascalientes, Mexico. This was a retrospective and observational study from a single center. Culture specimens were categorized according to their source such as blood, urine, respiratory, or biopsy. In case of a relapse, only the first bacterial isolation was considered. All the isolates were positive for *P. aeruginosa* by biochemical testing. As a confirmatory test, all the strains were confirmed by PCR as *Pseudomonas* sp., using the primers of PG-GS-F and PG-GS-R that mark the location 95–113 and 693–712 of the 16SrDNA sequence in *Pseudomonas* sp., as previously described [33]. *P. aeruginosa* ATCC 27853 was used as positive control and water as negative control. Primers used are listed in Appendix A.

### 2.2. Ethics Statement

The Centenario Hospital Miguel Hidalgo’s Ethics Committee approved this study on 16 January 2023, with the assigned number CEI-CI/008/23.

### 2.3. Data Collection

Clinical data were obtained from Centenario Hospital Miguel Hidalgo’s electronic clinical record. The repository was used to obtain clinical settings at the time of the culture, microbiology data, comorbidities, length of hospital stay from the day of the index culture (LOS), and complications of patients such as septic shock, mechanical ventilation, and previous antibiotic exposure (any empiric antibiotic prescribed 90 days before the positive culture).

### 2.4. Antibiotic Susceptibility Testing

All the strains were tested for antimicrobial susceptibility using the Vitek-2 analysis system (bioMérieux, Salt Lake City, UT, USA), according to the manufacturer’s instructions. The antibiotics tested were amikacin, cefepime, ceftazidime, ceftriaxone, ciprofloxacin, gentamicin, imipenem, meropenem, doripenem, piperacillin/tazobactam, colistin, and tigecycline. The MIC breakpoint values for each antibiotic are shown in Appendix A, CLSI 2020 [34]. However, not all the strains were tested against all the antimicrobial agents since not all the antibiotics were available at the hospital at the time of isolation. Bacteria strains were classified as carbapenem-resistant *P. aeruginosa* (*N* = 52) when the bacteria were non-susceptible to imipenem, meropenem, or doripenem [34]. Strains displaying minimum inhibitory concentration (MIC) values of ≥8 µg/mL for carbapenems were considered resistant. *Escherichia coli* ATCC 25922 and *P*. *aeruginosa* ATCC 27853 were used as quality-control strains. The results were interpreted as susceptible (S), intermediate (I), or resistant (R). The isolates were defined as MDR, extensively drug-resistant (XDR), or pan drug-resistant (PDR) using the clinical laboratory antimicrobial susceptibility testing results and according to a consensus definition [35]. Only carbapenem-resistant *P. aeruginosa* isolates were included in the study.

### 2.5. Screening and Identification of PMQR and Carbapenemase-Encoding Genes

For DNA extraction, bacteria were cultured on blood agar for 24 h at 37 °C. The culture was transferred into a sterile Broth Heath Infusion, BHI (BD, Le Pont de Claix, France) 5 mL tube, followed by incubation at 37 °C overnight. One point five mL of the bacterial culture were transferred to microcentrifuge tubes and the cells were pelleted at 12,000 rpm for 5 min. The supernatant was discarded, and the pellet was used for DNA isolation as described by Sambrook and Russell 2001 [36]. The DNA was stored at −20 °C until its use.

First, all the strains were analyzed for the genes encoding the following: (1) serino-β-lactamases (*bla*_GES_ and *bla*_KPC_); (2) metallo-β-lactamases (*bla*_IMP_, *bla*_VIM_, and *bla*_NDM_), and (3) oxacillinases (*bla*_OXA-23_, *bla*_OXA-48_, *bla*_OXA-51_, and *bla*_OXA-1_) [37,38]. The amplification programs were performed at 94 °C for 5 min, followed by 35 cycles. Each cycle consisted of 94 °C for 30 s, with various annealing conditions (Appendix A) for 45 s, and 72 °C for 1 min, with a final extension step of 72 °C for 10 min. Then, a screening for PMQR genes was performed as previously described [39,40,41,42,43]. The screening includes the detection of *qnrA*, *qnrB*, *qnrS*, *qnrC,* and *qnrD*, as well as the *oqxA* and *acc-(6′)-lb* genes. J53pMG252 strain was used as *qnrA* positive control, J53pMG298 as *qnrB* control, J53pMG306 as *qnrS* control, and *Salmonella* SA20042859 as a positive control to *acc-(6′)-lb*. In some cases, previous clinical strains carrying the different target genes were used as positive controls, and water as a negative control. The colistin resistance (*mcr*-1) gene was also tested. The amplification programs were performed at 94 °C for 5 min, followed by 35 cycles. Each cycle consisted of 94 °C for 30 s, 55°C for 45 s, and 72 °C for 30 s, with a final extension step of 72 °C for 10 min. Primers used are listed in Appendix A. The amplification products were visualized in agarose gel at 1.5%.

### 2.6. Statistical Analysis

Statistical data analysis was performed using GraphPad Prim Software (Boston, MA, USA, version 10.0.3). The results were described as descriptive statistics in terms of relative frequency. Comparisons among groups were made using the chi-square, Fisher’s exact test for the categorical data, or the Mann–Whitney U test for the continuous data. Univariate analysis of risk factors for 30-day mortality after positive carbapenem-resistant *P. aeruginosa* culture was carried out. *p*-values less than 0.05 were considered statistically significant.

## 3. Results

### 3.1. Identification of the Isolates and Clinical Characteristics

Based on a standard bacteriological test, 52 clinical strains were collected overall. All the isolates were positive for *P. aeruginosa* by biochemical and PCR testing. The clinical characteristics of the patients and the results of the univariate analysis of risk factors for 30-day mortality are shown in Table 1. Initial diagnosis or service of the patients include the following: viral pneumonia (20 isolates, 38.5%), Coronavirus (COVID) (6 isolates, 11.5%), acute respiratory failure (3 isolates, 5.8%), septic shock (2 isolates, 3.8%), kidney stone (3 isolates, 5.8%), woman surgery (2 isolates, 3.8%), septic arthritis (1 isolate, 1.9%), chronic obstructive pulmonary disease (1 isolate, 1.9%), multiple burns (1 isolate, 1.9%), general therapy (5 isolates, 9.6%), encephalopathy (1 isolate, 1.9%), and other conditions (7 isolates, 13.5%). The cases included 28 (53.8%) males and 24 (46.2%) females. The isolates were distributed among all the age groups, with 19 (36.5%) isolates belonging to the age category between 20 and 49 years old, 16 (30.8%) isolates between 50 and 64 years old, and 17 (32.7%) isolates ≥ 65 years old. Most isolates were collected from the respiratory tract (bronchial discharge culture, 34 isolates, 63.5%), followed by urine culture (13 isolates, 25%). Septic shock (OR = 0.19; 95% CI 0.04–0.79, *p* = 0.023) and infection with XDR bacteria (OR = 0.17; 95% CI 0.03–0.88, *p* = 0.035) were independent risk factors for 30-day mortality after the culture. Furthermore, 12 strains (23.1%) were isolated from patients with co-infections, including co-infection with *Acinetobacter baumannii* (4 isolates, 7.7%), *Klebsiella pneumoniae* (3 isolates, 5.8%), *Staphylococcus aureus* (1 isolate, 1.9%), *Chyseobacterium* (F.) *indologenes* (2 isolates, 3.8%), *Candida albicans* (1 isolate, 1.9%), and *Candida tropicals* (1 isolate, 1.9%).

### 3.2. Antimicrobial Susceptibility

Colistin was the most active agent tested for *P. aeruginosa* infection, with 81.8% susceptibility (Table 2). All the strains exhibited resistance to imipenem (MIC values ranging from 8 to ≥16 µg/mL), meropenem (MIC values ranging from 8 to ≥16 µg/mL), doripenem (MIC values ≥ 8 µg/mL), ceftriaxone (MIC values ranging from 32 to ≥64 µg/mL), and tigecycline (MIC values ≥ 8 µg/mL). In addition, 15 of 52 isolates (28.8%) were categorized as MDR, 30 isolates (57.7%) as XDR, and seven isolates (13.5%) as PDR. Interestingly, all XDR strains were colistin-susceptible. Individual results are described in Appendix A.

To determine if there were some differences in the antibiotic-resistant profiles of the isolates based on the susceptibility to quinolones, it was compared to ciprofloxacin-resistant isolates vs. ciprofloxacin-susceptible isolates (Table 3). Antibiotic resistance rates between the two groups were significantly different. Notably, ciprofloxacin-susceptible isolates had higher rates of susceptibility to several antibiotics, including cefepime (50%, *p* = 0.0018), ceftazidime (33.3%, *p* = 0.0046), amikacin (100%, *p* < 0.0001), and gentamicin (100%, *p* < 0.0001). Based on these findings, it is suggested that ciprofloxacin may be a viable treatment option for this particular phenotype.

### 3.3. Occurrence of Carbapenemase Encoding-Genes

Out of 52 isolates, 35 (67.3%) were positive for at least one carbapenemase-encoding gene tested. Two isolates (3.8%) were positive for *bla*_IMP_; three (5.8%) were positive for *bla*_KPC_ and *bla*_NDM_; four (7.7%) were positive for *bla*_OXA-48_, and eight (15.4%) were positive for *bla*_OXA-1_ and *bla*_VIM_. *bla*_GES_ was detected only in one isolate (1.9%), and *bla*_OXA-51_ was the most frequently carbapenemase-encoding gene found within 22 isolates (42.3%). None of the isolates tested carried *bla*_OXA-23_. Seventeen isolates (32.7%) were negative for carbapenemase-encoding genes, suggesting that other mechanisms of resistance to carbapenems are important for the strains isolated at the hospital. Moreover, 38.5% (20 isolates) were positive for only one carbapenemase-encoding gene, 26.9% (14 isolates) co-carrying two different genes, and 1.9% (1 isolate) co-carrying three different genes (*bla*_OXA-51_, *bla*_OXA-1_, and *bla*_VIM_). The distribution of carbapenemase-encoding genes found among isolates is shown in Table 4.

### 3.4. Screening for Plasmid-Mediated Colistin-Resistant mcr-1 Gene

Only one isolate (1.9%) was positive for *mcr-1* (PHH12 isolate, Appendix A). This strain also carried *bla*_OXA-48_ and *qnrS* genes, was colistin intermediate with MIC values ≤ 0.5 µg/mL, ciprofloxacin-resistant (MIC ≥ 4 µg/mL), doripenem resistant (MIC ≥ 8 µg/mL), imipenem resistant (MIC ≥ 16 µg/mL), and meropenem resistant (MIC ≥ 16 µg/mL). Furthermore, this strain was susceptible to amikacin (MIC = 8). Interestingly, none of the three isolates resistant to colistin (MICs values of ≥16 µg/m and 4 µg/mL) were positive for the *mcr-1* gene, indicating the presence of other mechanisms of resistance such as chromosome-encoded mutations.

### 3.5. Presence of PMQR Determinants

Among the 52 isolates, 18 (34.6%) isolates were positive for the *qnr* genes, 24 (46.2%) isolates were positive for the *oqxA* gene, and 13 (25%) for the *aac-(6′)-lb* gene (Table 4). The results indicated that *oqxA* was the most prevalent quinolone-resistance gene, followed by the *aac-(6′)-lb* gene. In the case of *qnr* genes, *qnrS* was the most frequent gene (9 isolates, 17.3%), followed by *qnrC* (7 isolates, 13.5%) and *qnrB* (2 isolates, 3.8%). *qnrD* and *qnrA* were not found in the isolates (Appendix A).

### 3.6. Co-Occurrence of Carbapenemase-Encoding Genes and PMQR Determinants

The co-existence of carbapenemase-encoding and quinolone-resistant genes was found among CIP-resistant and CIP-sensible isolates (Table 4). The patterns of co-existence were recognized as follows: *bla*_OXA-51_ co-carrying *oqxA* (11 isolates, 21.1%); *bla*_OXA-51_ co-carrying *aac-(6´)-lb* (6 isolates, 11.5%); *bla*_VIM_ co-carrying *aac-(6′)-lb* (2 isolates, 3.8%); *bla*_KPC_ co-carrying *oqxA* (3 isolates, 5.8%), with two of them also carrying *qnrS*; and *bla*_OXA-48_ co-carrying *qnrS* (3 isolates, 5.8%), with two of them also carrying *oqxA*. However, it seems that the co-existence is unrelated to a specific pattern of ciprofloxacin-resistant determinants. No significant differences were found between PMQR and ciprofloxacin-resistant and sensitive isolates.

## 4. Discussion

The incidence of plasmid-mediated quinolone-resistant determinants and carbapenemase-encoding genes simultaneously in *P. aeruginosa* has been analyzed in limited studies. In this study, the prevalence of PMQR and carbapenemase-encoding genes occurring at the same time in *P. aeruginosa* strains isolated from clinical patients in Aguascalientes, Mexico was reported. Fifty-two strains were investigated. The highest proportion of carbapenem-resistant *P. aeruginosa* was isolated from respiratory tract samples. This was in concordant with previous studies since carbapenem resistance is directly influenced by the culture site, being higher in respiratory infections than in bloodstream infections [44]. Additionally, urine cultures had a great proportion of carbapenem-resistant *P. aeruginosa* isolates in this study. Moreover, we found that septic shock was an independent risk factor for 30-day mortality as well as extensively drug-resistant *P. aeruginosa* (XDR-PA), in agreement with previous studies [45,46,47].

Higher resistance levels to all tested antibiotics were identified in the study by Nieto-Saucedo et al. [13]. In the comparative analysis, resistance rates were as follows: amikacin (24% vs. 73.1%), gentamicin (60% vs. 75%), ceftazidime (52% vs. 83.3%), cefepime (44% vs. 76%), ciprofloxacin (44% vs. 88.4%), colistin (0% vs. 6.8%), and piperacillin/tazobactam (36% vs. 71.8%). This could be due to geographic variation, healthcare practices, and the genetic mechanisms underlying resistance. Conversely, the study by Martinez-Zavaleta et al. [48], also from Mexico, reported similar resistance rates in comparison to our findings, showing resistance levels for amikacin (50% vs. 73.1%), gentamicin (70.8% vs. 75%), ceftazidime (73.9% vs. 83.3%), cefepime (76.5% vs. 76%), colistin (9.9% vs. 6.8%), and piperacillin/tazobactam (57.3% vs. 71.8%). Furthermore, a study in Nigeria [49] reported resistance rates of approximately 75% for ceftazidime, cefepime, ciprofloxacin, and piperacillin/tazobactam in CRPA. Another study in Japan found that 47.1% of the CRPA isolates were resistant to piperacillin-tazobactam and 73.4% to ciprofloxacin [50], which aligns with the antimicrobial susceptibility found in this study.

Additionally, colistin (polymyxin E) was the most active agent tested, with less than seven percent of the isolates resistant, suggesting its use as a better alternative to treat these infections. The prevalence of colistin resistance was higher than one reported in a previous study in Mexico (1.65% of prevalence) [51]. Even though colistin seems to be the best option for treatment due to a lower resistance rate, this antibiotic could cause harmful effects on human health, and precautions must be taken regarding its use. There were notable variations in antibiotic resistance rates between the ciprofloxacin-resistant and ciprofloxacin-susceptible isolates, which differed from a previous study [15]. Furthermore, the ciprofloxacin-susceptible isolates exhibited higher susceptibility to aminoglycosides, ceftazidime, and cefepime, suggesting a viable alternative for this type of phenotype. Similarly, in a previous study [45], all XDR-*P. aeruginosa* isolates were susceptible to colistin, suggesting its use as a better alternative treatment.

This study included only the CRPA isolates and the first eligible culture episode for each patient. This approach was taken to minimize the chances of including the same isolate multiple times, as some patients experienced two or three reinfections during the study period. Moreover, the period of the study (2019 to 2023), the diversity of the specimen types (blood, biopsy, respiratory tract, and urine), and the genes found in the different isolates suggest that they represent distinct bacterial strains.

Among carbapenemase-encoding genes, Nieto-Saucedo et al. [13] found the *bla*_IMP-75_ gene as the most common carbapenemase-encoding gene for *P. aeruginosa*, contrasting with our study since we detected only two isolates carrying *bla*_IMP_ genes. The genes *bla*_GES_ and *bla*_VIM_ were also found in *P. aeruginosa* [13,32]. In our study, *bla*_GES_ and *bla*_VIM_ were also detected in the isolates, with *bla*_VIM_ and *bla*_OXA-51_ being the most frequently detected carbapenemase-encoding genes, which agrees with previous studies in Mexico, where *bla*_VIM_ resulted in the primary gene reported in the same bacteria [11]. Detection of *bla*_OXA-51_ is rare for *P. aeruginosa* since this gene encoding Ambler class D oxacillinase (OXA)-like carbapenemase, *bla*_OXA-51_ is naturally detected in *A. baumannii* and is related in resistance to oxacillin, cephalosporins, and carbapenems [52,53]. However, in the last years, it has been emerging in *P. aeruginosa* [54], as well as other oxacillinase-type enzymes such as *bla*_OXA-48_ [55,56]. Moreover, we included various types of hospital-acquired infections caused by *P. aeruginosa* such as monomicrobial and polymicrobial infections. Indeed, several strains were isolated from patients with co-infection by *A. baumannii*, thus facilitating the spread of resistant determinants through horizontal gene transfer, as well as influencing the treatment responses and patient outcomes.

Carbapenemase-encoding genes were not amplified in thirty-two percent of the isolates. Therefore, the production of carbapenemases may not be the primary mechanism of resistance in *P. aeruginosa* strains circulating in the hospital, which is closer to those described by Garza-Ramos et al. [11], where 47% of the isolates were negative for the presence of the carbapenemase-encoding genes tested. Other mechanisms may include alteration or lack of porin OprD which has been related to reduced susceptibility to carbapenems [57], overexpression of efflux pumps, mutations in AmpC and its regulatory genes, and plasmid-encoded AmpC. Nevertheless, detecting MBL-producing strains is of great importance since options for *P. aeruginosa* are still limited. Lately, the treatment suggested for MBL-carbapenemase includes cefiderocol, fosfomycin, high-dose amikacin, and a synergic combination of colistin with fosfomycin or aminoglycoside, or the combination of ceftolozane-tazobactam with amikacin or fosfomycin [58,59]. Meanwhile, ceftazidime-avibactam could be usable for GES-carbapenemase *P. aeruginosa* [58], KPC and OXA-48 producers; ceftazidime-avibactam plus aztreonam seems to be a reliable option for MLBs producers [31].

Only one strain tested carried the *mcr-1* gene. Despite their harmful effects on human health, polymyxins remain a last resort against Gram-negative MDR infections, especially in carbapenem-resistant bacteria. The *mcr* genes are of great concern due to their high potential for horizontal propagation. Among the variants described, *mcr-1* is the most prevalent within Mexico [11] and globally [60]. Previous reports have also revealed the emergence of *mcr-1* in *P. aeruginosa* [61,62]. In Mexico, *mcr-1* has also been reported in *Klebsiella pneumoneae* [11] and *Escherichia coli* [51]; meanwhile, *mcr-2* has been detected in *Enterobacter cloacae* [13]. The horizontal propagation of the *mcr*-1 gene into carbapenem-resistant bacteria could severely limit antimicrobial treatment options. Thus, it is essential to strengthen our surveillance of bacteria exhibiting this resistance mechanism.

Among fluoroquinolone resistance genes, there were found frequencies of 46.2% for *oqxA*, 25% for *aac-(6′)-lb*, and 34.6% for *qnr* genes (17.3% for *qnrS*, 13.5% for *qnrC*, and 3.8% for *qnrB*). The *qnrD* and *qnrA* genes were not detected. This agrees with Venkataramana et al. [19], who found that plasmid-meditated resistance *acc-(6′)-lb-cr* is primarily responsible for mediating fluoroquinolone resistance in clinical isolates of *P. aeruginosa*. However, higher frequency was found within 77.6% of the isolates testing positive for this gene. The authors also found the genes *qnrB* (14.1%), *qnrS* (14.1%), and *oqxAB* (3.5%) in the isolates. Similar results were detected by Abdelrahim et al., who showed a frequency of 77.3% for *acc-(6′)-lb-cr* gene in *P. aeruginosa* [63]. In our study, several ciprofloxacin-resistant isolates were negative for the PMQR tested. This might be attributable to the major mechanism for fluoroquinolone resistance in *P. aeruginosa*, the mutations in the DNA gyrase encoded by the *gyrA* and *gyrB* genes, as well as in the topoisomerase IV encoded by the *parC* and *parE* genes [64].

Interestingly, in our study, *oqxA* has the highest frequency (46.2%), which encodes for a multi-drug efflux pump belonging to the resistance-nodulation-division family (RND) [14], implying its contribution to the reduced susceptibility to quinolones in the clinical isolates of the region. In the study by Andres et al. [65], *oqxA* and *oqxB* were found in isolates with a wide range of MIC values for ciprofloxacin, including some that were susceptible. Moreover, Goudarzi et al. [66] highlighted that these genes were detected in both susceptible and resistant isolates. This finding is supported by Agyepong et al. [67], who observed that *oqxA* and *oqxB* were present together in susceptible and resistant ciprofloxacin isolates. This agrees with our study, where several strains carrying *oqxA* were ciprofloxacin-sensitive isolates, and the resistant ones have MIC values ≥ 4 µg/mL, suggesting that while *oqxA* is associated with resistance, it can also coexist with susceptibility or low-level-resistant isolates. Furthermore, the detection of *oqxA* gene is worrisome since the OqxAB multi-drug efflux pump is linked to low to intermediate resistance to other antibiotics such as quinoxalines, quinolones, tigecycline, nitrofurantoin, chloramphenicol, several detergents and disinfectants [68], and the high level of expression of this efflux pump is associated to high resistant to ciprofloxacin [69]. Additionally, the presence of *aac(6′)-lb-cr* gene enables the selection of highly ciprofloxacin-resistant chromosomal mutants. It converts the low-level fluoroquinolone resistance mediated by this enzyme into high-level resistance when present alongside Qnr proteins [63].

Like earlier studies [19,70], the *qnrA* gene was not detected in any of the isolates of *P. aeruginosa* in this study. In contrast to our study, Lopez-Garcia et al. [32] did not find the genes *bla*_VIM_, *bla*_NDM_, *bla*_KPC_, *qnrA*, *qnrB*, *qnrS* or *aac-(6′)-Ib-cr* in MDR *P. aeruginosa* strains. Regardless, *bla*_IMP_ (41%), *bla*_GES_ (49%), *bla*_OXA-2_ (85%), and *bla*_OXA-50_ (100%) were found. Another study in Iran [15] found higher prevalence rates of *qnr* genes compared to our study: *qnrA* (25.8% vs. 0%), *qnrB* (29.2% vs. 3.8%), and *qnrS* (20.8% vs. 17.3%); however, *qnrD* was also not detected as in this study.

Although some studies were carried out to detect PMQR in *P. aeruginosa*, recent reports suggest a current increase in the mobile genetic pool in *P. aeruginosa* which could provide multiple resistance to the most clinically used antibiotics [71]. Indeed, *qnrS1* was detected alongside the β-lactamase gene *bla*_TEM-1_ [72], while *qnrVC* was detected with the *bla*_VIM-2_ gene as part of class 1 integron [73]. Elena et al. [22] found the simultaneous presence of *bla*_VIM-11_ and the PMQR *qnrS1* in the same strain located in different plasmids. Other studies have shown evidence of *qnrVC1* and *bla*_NDM-1_ in the high-risk *P. aeruginosa* ST773 clone reported worldwide as MDR [24]. Lopez-Garcia et al. [32] found *bla*_IMP-18_ + *aacA7* + *bla*_IMP-62_ + *qacH* + *aacA4* + *aadA1* + *bla*_OXA-2_ in two strain isolates from the same patient. Similarly, the co-existence of resistance genes in different combinations in *P. aeruginosa* has also been previously reported with multiple ESBL genes and carbapenemase-encoding genes (CTX-M-1, NDM-1, and OXA-48), with the addition of *acc-(6′)-lb* [74].

In this study, it was found that PMQR was present on resistant and susceptible ciprofloxacin isolates, indicating a possible future increase in quinolone resistance. Several isolates carried carbapenemase-encoding genes, and PMQR determinants simultaneously were found, indicating the dissemination of PMQR on carbapenem-resistant strains. Figure 1 describes the presence of both genetic elements in *P. aeruginosa*. OXA-type carbapenemase-encoding genes were found in combination with efflux pumps and aminoglycoside variants (*bla*_OXA-51_ + *oqxA* + *aac-(6′)-lb*) as well as serin-based carbapenemase-encoding genes with *qnr*-encoding variants and efflux pump genes (*bla*_KPC_ + *qnrB*, *qnrC*, *qnrS* + *oqxA*; *bla*_GES_ + *qnrB*, *qnrS* + *oqxA*). The co-carriage of carbapenemase-encoding genes and PMQR determinants might suggest the presence of circulating plasmids that carry these resistance genes [63]. *P. aeruginosa* is a highly diverse pathogen that can adapt to its environment. The rapid development of antimicrobial resistance in this bacterium may be attributed to the excessive and inappropriate use of antibiotics. This creates selective pressure which can lead to mutations, the acquisition of resistance genes through horizontal gene transfer, the overexpression of beta-lactamases, and the proliferation of antibiotic resistance [75]. While PMQR determinants by themselves generally produce low levels of fluoroquinolone resistance, this plays a significant role in the emergence of clinical resistance to ciprofloxacin. This allows the bacteria to grow at clinically relevant concentrations of quinolones, potentially leading to treatment failures.

The frequency of PMQR genes in clinical isolates of carbapenem-resistant *P. aeruginosa* is worrisome since it enables their spread to other bacterial species by horizontal gene transfer. It is important to emphasize that understanding the mechanisms of carbapenem resistance in clinical isolates of *P. aeruginosa* is crucial in determining the most effective treatment strategy.

Our study has some limitations. Firstly, all the strains were not tested with all antibiotics because not all antibiotics were available at the time of the isolation. Secondly, the study involved patients in one hospital center. To increase the statistical power of the results, it would be necessary in the future to include a higher number of samples in a larger multicenter study to improve the generalizability of the findings. However, although this study is limited to a single center, it aims to guide the rational prescription of antibiotics in a regional context. Finally, although all isolates come from different patients, the study did not assess the genetic relationships among the resistant strains.

## 5. Conclusions

To the best of our knowledge, this is the first report in Mexico of PMQR genes in carbapenem-resistant *P. aeruginosa*. The co-existence of PMQR with carbapenemase-encoding genes in carbapenem-resistant *P. aeruginosa* is infrequent, though its occurrence could complicate infection treatments and increase resistance through horizontal gene transfer. Routine screenings are advisable as an infection control measure. This study aims to guide the rational prescription of antibiotics in a regional context.

## Figures and Tables

**Figure 1 pathogens-13-00992-f001:**
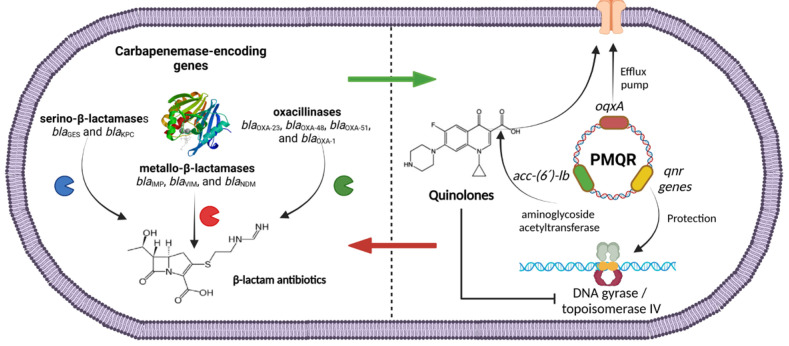
Plasmid-mediated quinolone-resistance genes (PMQR) and carbapenemase-encoding genes found in CRPA isolates in this study.

**Table 1 pathogens-13-00992-t001:** Univariate analysis of risk factors for the 30-day mortality in patients with carbapenem-resistant *P. aeruginosa*.

Variables	Total *n*, (%)	Death 30-Day After Culture (%), [*n *= 13]	Non-Death 30 Days After Culture (%), [*n* = 39]	*^a^* Odds Ratios (95% CI)	*^b^* *p* Value
Clinical characteristics					
Female	24 (46.2)	5 (38.5)	19 (48.7)	Reference	
Male	28 (53.8)	8 (61.5)	20 (51.3)	0.66 (0.18–2.37)	0.522
Age category					
20–49	19 (36.5)	4 (30.8)	15 (38.5)	1.41 (0.37–5.39)	0.619
50–64	16 (30.8)	4 (30.8)	12 (30.8)	1 (0.26–3.9)	1.000
≥65	17 (32.7)	5 (38.5)	12 (30.8)	1.41 (0.38–5.2)	0.609
Specimen type					
Respiratory	34 (65.4)	10 (76.9)	24 (61.5)	0.48 (0.11–2.03)	0.319
Urine	13 (25.0)	1 (7.7)	12 (30.8)	0.19 (0.02–1.61)	0.127
Blood	2 (3.8)	1 (7.7)	1 (2.6)	0.32 (0.02–5.44)	0.427
Biopsy	3 (5.8)	1 (7.7)	2 (5.1)	0.65 (0.05–7.8)	0.733
Co-morbidities					
Diabetes mellitus	27 (51.9)	8 (61.5)	19 (48.7)	0.59 (0.16–2.14)	0.425
Systemic arterial hypertension	30 (57.7)	8 (61.5)	22 (56.4)	1.24 (0.34–4.47)	0.746
Chronic kidney disease	3 (5.8)	0 (0)	3 (7.7)	-	-
Congestive heart failure	4 (7.7)	2 (15.4)	2 (5.1)	0.3 (0.04–2.36)	0.251
Hypothyroidism	7 (13.5)	0 (0)	7 (17.9)	-	-
Obesity	33 (63.5)	8 (61.5)	25 (64.1)	1.12 (0.31–4.07)	0.868
COVID-19	35 (67.3)	10 (76.9)	25 (64.1)	0.54 (0.13–2.28)	0.398
Charlson Comorbidity Index					
<3	28 (53.8)	4 (30.8)	24 (61.5)	Reference	
≥3	24 (46.2)	9 (69.2)	15 (38.5)	0.8 (0.54–1.18)	0.1063
Complication					
Mechanical ventilation	43 (82.7)	12 (92.3)	31 (79.5)	0.32 (0.04–2.87)	0.310
Pneumonia associated with mechanical ventilation	43 (82.7)	12 (92.3)	31 (79.5)	0.32 (0.04–2.87)	0.310
Tracheostomy	28 (53.8)	6 (46.2)	22 (56.4)	1.51(0.43–5.33)	0.522
Septic shock	25 (48.1)	10 (76.9)	15 (38.5)	0.19 (0.04–0.79)	**0.023**
ICU	21 (40.4)	6 (46.2)	15 (38.5)	0.73 (0.21–2.59)	0.625
Antibiotic exposure in the previous 90 days					
Any antibiotics					
Yes	40 (76.9)	10 (76.9)	30 (76.9)	1.0 (0.23–4.44)	1.000
No	12 (23.1)	3 (23.1)	9 (23.1)		
Adequate treatment					
Yes	34 (65.4)	8 (61.5)	26 (66.7)	0.8 (0.22–2.94)	0.737
No	18 (34.6)	5 (38.5)	13 (33.3)		
Antimicrobial resistance					
*^b^* MDR	15 (28.8)	1 (7.7)	14 (35.9)	0.15 (0.02–1.27)	0.081
XDR	30 (57.7)	11 (84.6)	19 (48.7)	0.17 (0.03–0.88)	*^c^* **0.035**
PDR	7 (13.5)	1 (7.7)	6 (15.4)	2.18 (0.24–20.44)	0.491

***^a^*** CI: confidence interval; *^b^* Statistically significant *p*-values are present in bold (*p* ≤ 0.05). MDR: multidrug-resistant; XDR: extensively drug-resistant; PDR: pandrug-resistant. *^c^* Mann–Whitney U value.

**Table 2 pathogens-13-00992-t002:** Antimicrobial susceptibility of *P. aeruginosa* isolates (N = 52).

Antimicrobial Class	Antimicrobial Agent	Susceptibility (%)		
		Susceptible	Intermediate	Resistant
Quinolones	Ciprofloxacin (*n* = 52)	6 (11.5)	0 (0)	46 (88.5)
Carbapenems	Imipenem (*n* = 44)	0 (0)	0 (0)	44 (100)
	Meropenem (*n* = 51)	0 (0)	0 (0)	51 (100)
	Doripenem (*n* = 44)	0 (0)	0 (0)	44 (100)
Polymyxin E	Colistin (*n* = 44)	36 (81.8)	5 (11.4)	3 (6.8)
Glycylcline	Tigecycline (*n* = 42)	0 (0)	0 (0)	42 (100)
Penicillin and beta-lactamase inhibitors	Piperacillin/tazobactam (*n* = 39)	3 (7.7)	8 (20.5)	28 (71.8)
Cephalosporins	Cefepime (*n* = 52)	7 (13.5)	5 (9.6)	40 (76.9)
	Ceftazidime (*n* = 48)	4 (8.3)	4 (8.3)	40 (83.3)
	Ceftriaxone (*n* = 48)	0 (0)	0 (0)	48 (100)
Aminoglycosides	Amikacin (*n* = 52)	14 (26.9)	0 (0)	38 (73.1)
	Gentamicin (*n* = 52)	9 (17.3)	4 (7.7)	39 (75.0)

**Table 3 pathogens-13-00992-t003:** Antibiotic resistance rates of ciprofloxacin-resistant and ciprofloxacin-susceptible *P. aeruginosa*.

*^a^* Antibiotics	Ciprofloxacin-Resistant			Ciprofloxacin-Susceptible			* *p*-Value
	R, *n* (%)	I, *n* (%)	S, *n* (%)	R, *n* (%)	I, *n* (%)	S, *n* (%)	
Colistin (*n* = 44) CIP-R (*n* = 38), CIP-S (*n* = 6)	2 (5.3)	4 (10.5)	32 (84.2)	1 (16.65)	1 (16.65)	4 (66.7)	0.2968
Tigecycline (*n* = 42) CIP-R (*n* = 36), CIP-S (*n* = 6)	36 (100)	0 (0)	0 (0)	6 (100)	0 (0)	0 (0)	>0.9999
Piperacillin/tazobactam (*n* = 39) CIP-R (*n* = 34), CIP-S (*n* = 5)	26 (76.5)	5 (14.7)	3 (8.8)	2 (40)	3 (60)	0 (0)	0.0871
Cefepime (*n* = 52) CIP-R (*n* = 46), CIP-S (*n* = 6)	39 (84.8)	3 (6.5)	4 (8.7)	1 (16.7)	2 (33.3)	3 (50)	**0.0018**
Ceftazidime (*n* = 48) CIP-R (*n* = 42), CIP-S (*n* = 6)	38 (90.4)	2 (4.8)	2 (4.8)	2 (33.33)	2 (33.33)	2 (33.33)	**0.0046**
Ceftriaxone (*n* = 48) CIP-R (*n* = 43), CIP-S (*n* = 5)	43 (100)	0 (0)	0 (0)	5 (100)	0 (0)	0 (0)	>0.9999
Amikacin (*n* = 52) CIP-R (*n* = 46), CIP-S (*n* = 6)	40 (87.0)	0 (0)	6 (13.0)	0 (0)	0 (0)	6 (100)	**<0.0001**
Gentamicin (*n* = 52) CIP-R (*n* = 46), CIP-S (*n* = 6)	39 (84.8)	4 (8.7)	3 (6.5)	0 (0)	0 (0)	6 (100)	**<0.0001**

* Statistically significant *p*-values are presented in bold (*p* ≤ 0.05). The results display the total number of CIP-R (ciprofloxacin-resistant) and CIP-S (ciprofloxacin-sensitive) isolates tested for each antibiotic separately. *^a^* Not all the susceptibility test results were available. Results are presented as R: resistant, I: Intermediate, and S: susceptible, isolate number/total (%).

**Table 4 pathogens-13-00992-t004:** Distribution of carbapenem-encoding genes and PMQR determinants in *Pseudomonas aeruginosa isolates* (N = 52).

Carbapenemase-Encoding Genes			PMQR Genes			PMCR ^a^	CIP-Susceptibility ^b^		Total Prevalence (N = 52)
MBL	OXA	SBL	Qnr Variants	Efflux Pumps	Aminoglycoside Variant	*mcr-1*	R ^c^ *n* = 46, (%)	S ^d^ *n* = 6, (%)	
*bla* _IMP_	*bla* _OXA-51_		*qnrC*	*oqxA*	*aac-(6´)-lb*		1 (2.2)	0 (0.0)	1 (1.9)
*bla* _IMP_	*bla* _OXA-51_						1 (2.2)	0 (0.0)	1 (1.9)
*bla* _NDM_	*bla* _OXA-1_				*aac-(6´)-lb*		1 (2.2)	0 (0.0)	1 (1.9)
*bla* _NDM_	*bla* _OXA-1_						1 (2.2)	0 (0.0)	1 (1.9)
*bla* _NDM_				*oqxA*			1 (2.2)	0 (0.0)	1 (1.9)
*bla* _VIM_	*bla* _OXA-51_				*aac-(6´)-lb*		1 (2.2)	0 (0.0)	1 (1.9)
*bla* _VIM_	*bla* _OXA-51_						1 (2.2)	0 (0.0)	1 (1.9)
*bla* _VIM_	*bla* _OXA-51_			*oqxA*			0 (0.0)	1 (16.7)	1 (1.9)
*bla* _VIM_		*bla* _KPC_	*qnrB*, *qnrC*, *qnrS*	*oqxA*			1 (2.2)	0 (0.0)	1 (1.9)
*bla* _VIM_	*bla* _OXA-48_		*qnrS*			*mcr-1*	1 (2.2)	0 (0.0)	1 (1.9)
*bla* _VIM_	*bla*_OXA-51_, *bla*_OXA-1_						1 (2.2)	0 (0.0)	1 (1.9)
*bla* _VIM_					*aac-(6´)-lb*		1 (2.2)	0 (0.0)	1 (1.9)
*bla* _VIM_							0 (0.0)	1 (16.7)	1 (1.9)
	*bla* _OXA-1_						2 (4.3)	0 (0.0)	2 (3.8)
	*bla* _OXA-1_		*qnrS*		*aac-(6´)-lb*		1 (2.2)	0 (0.0)	1 (1.9)
		*bla* _GES_	*qnrB*, *qnrS*	*oqxA*			0 (0.0)	1 (16.7)	1 (1.9)
	*bla* _OXA-48_	*bla* _KPC_	*qnrS*	*oqxA*			0 (0.0)	1 (16.7)	1 (1.9)
	*bla* _OXA-48_		*qnrS*	*oqxA*			1 (2.2)	0 (0.0)	1 (1.9)
	*bla* _OXA-51_	*bla* _KPC_		*oqxA*			1 (2.2)	0 (0.0)	1 (1.9)
	*bla*_OXA-51_, *bla*_OXA-1_			*oqxA*			1 (2.2)	0 (0.0)	1 (1.9)
	*bla*_OXA-51_, *bla*_OXA-1_			*oqxA*	*aac-(6´)-lb*		1 (2.2)	0 (0.0)	1 (1.9)
	*bla*_OXA-51_, *bla*_OXA-48_						1 (2.2)	0 (0.0)	1 (1.9)
	*bla* _OXA-51_				*aac-(6´)-lb*		2 (4.3)	0 (0.0)	2 (3.8)
	*bla* _OXA-51_			*oqxA*			4 (8.7)	0 (0.0)	4 (7.7)
	*bla* _OXA-51_						4 (8.7)	0 (0.0)	4 (7.7)
	*bla* _OXA-51_		*qnrC*	*oqxA*			1 (2.2)	0 (0.0)	1 (1.9)
	*bla* _OXA-51_			*oqxA*	*aac-(6´)-lb*		1 (2.2)	0 (0.0)	1 (1.9)
			*qnrC*, *qnrS*	*oqxA*			2 (4.3)	1 (16.7)	3 (5.8)
			*qnrC*				1 (2.2)	0 (0.0)	1 (1.9)
				*oqxA*	*aac-(6´)-lb*		1 (2.2)	0 (0.0)	1 (1.9)
				*oqxA*			3 (6.5)	1 (16.7)	4 (7.7)
					*aac-(6´)-lb*		3 (6.5)	0 (0.0)	3 (5.8)
-	-	-	-	-	-		5 (10.9)	0 (0.0)	5 (9.6)

^a^ PMCR: plasmid-mediated colistin resistant, ^b^ CIP-susceptibility: ciprofloxacin susceptibility, ^c^ R: resistant, ^d^ S: susceptible. The hyphen indicates that there was no amplification of any carbapenemase-encoding genes or PMQR genes.

## Data Availability

Data are contained within the article.

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
