# Peer review of "Occurrence of Plasmid-Mediated Quinolone Resistance and Carbapenemase-Encoding Genes in Pseudomonas aeruginosa Isolates from Nosocomial Patients in Aguascalientes, Mexico"

_pathogens, 2024, doi:10.3390/pathogens13110992_

Round 1
Reviewer 1 Report
Comments and Suggestions for Authors
The authors have carried out a survey of 52 carbapenem-resistant isolates from nosocomial infections for their resistance to various antibiotics. They have also tested the isolates for the presence of b lactamase and PMQR genes that can contribute to resistance to carbapenems and ciprofloxacin. Their findings indicate that PMQR genes are present at significant frequencies in the isolates studied. The conclusions would be strengthened by evidence that the isolates studied were unrelated to each other, despite being from the same hospital; and by analysis of the frequency of PMQR genes in isolates that were carbapenem-sensitive.
1. All of the isolates described in the study were resistant to carbapenems. Does that mean that all isolates from the hospital over the time period were carbapenem resistant? Or were isolates that were sensitive to carbapenems excluded from the study? – this should be specified. Assuming the latter, what proportion of isolates from the hospital during the sampling period were carbapenem-resistant?
2. All of the isolates were from the same hospital. Were all of the isolates from different patients? As they were nosocomial infections, there is the possibility of a common source of infection which could cause identical bacteria to be isolated from different patients. The inadvertant inclusion of multiple samples of the same isolate(s) would affect the statistical analysis. Did the authors carry out any typing to show that all isolates were independent and unrelated? If not, the possibility that duplicates (or more) of the same isolate might have been included and the impact that would have on results needs to be discussed.
3. Antibiotic susceptibility testing – what were the resistance/ intermediate MIC threshholds for antibiotics other than carbapenems?
4. Table 1. What is “reference”? “Odd ratio” should presumably be “Odds ratio”, as the full term is used in the table heading it is not necessary to define “OR” in the footnote.
5. Line 212. What is “Non-blaOXA23”?
6. Table 3. The p-value describes (presumably) whether there is a significant difference between ciprofloxacin-resistant and -sensitive samples. As only carbapenem-resistant isolates were included in the study, it is meaningless to assess the difference between cipro-resistant and -sensitive isolates and the lines for imipenem, meropenem and doripenem can be deleted.
7. Fluoroquinolone resistance. Table 3 indicates that a lot of the ciprofloxacin-resistant isolates did not have any PMQR genes. Presumably these isolates are resistant due to mutations in gyrA or other genes associated with resistance. In addition, oqxA is present in ciprofloxacin-sensitive strains. The authors should discuss whether the isolates that they studied also contain mutations in gyrA or other genes, and the extent to which the PMQR genes contribute to the ciprofloxacin-resistance phenotype.
8. Lines 334-352, presence of PMQR and b lactamase genes in the same isolates. Do the authors think that the PMQR genes and b lactamase genes are present on the same genetic elements in the isolates in their study? Why do their results indicate “dissemination of PMQR genes on carbapenem-resistant strains”, why not dissemination of b lactamase genes on ciprofloxacin-resistant strains? This paragraph is something of a catalogue, documenting other studies – what is the take-home message?
9. In connection with that point, the study would be greatly strengthened by analysis of PMQR genes in carbapenem-sensitive isolates from the same hospital – is the frequency of PMQR genes lower in isolates that are carbapenem-sensitive but ciprofloxacin-resistant?
10. Why would understanding the mechanism of carbapenem-resistance be “invaluable to take measures and prevent the spread of emerging resistance mechanisms in hospital environments” (line 357)? – please explain.
Author Response
For research article
|
Response to Reviewer 1 Comments
|
||
|
1. Summary |
|
|
|
Thank you very much for taking the time to review this manuscript. Please find the detailed responses below and the corresponding revisions/corrections highlighted/in track changes in the re-submitted files. We appreciate all your valuable comments and suggestions, which have helped us to improve the quality of the manuscript. We considered carefully each of the comments and tried our best to answer and solve them. We have incorporated into the new version of the manuscript most of the reviewers’ suggestions, and the changes are highlighted within the manuscript. We sincerely hope that you will find our revised manuscript adequate for publication.
We break down the answers given to each reviewer below.
|
||
|
2. Questions for General Evaluation |
Reviewer’s Evaluation |
Response and Revisions |
|
Does the introduction provide sufficient background and include all relevant references? |
Yes/Can be improved/Must be improved/Not applicable |
We response in the point-by-point response letter. |
|
Are all the cited references relevant to the research? |
Yes/Can be improved/Must be improved/Not applicable |
|
|
Is the research design appropriate? |
Yes/Can be improved/Must be improved/Not applicable |
|
|
Are the methods adequately described? |
Yes/Can be improved/Must be improved/Not applicable |
|
|
Are the results clearly presented? |
Yes/Can be improved/Must be improved/Not applicable |
|
|
Are the conclusions supported by the results? |
Yes/Can be improved/Must be improved/Not applicable |
|
|
3. Point-by-point response to Comments and Suggestions for Authors |
||
|
1. Comment 1: All of the isolates described in the study were resistant to carbapenems. Does that mean that all isolates from the hospital over the time period were carbapenem resistant? Or were isolates that were sensitive to carbapenems excluded from the study? – this should be specified. Assuming the latter, what proportion of isolates from the hospital during the sampling period were carbapenem-resistant?
|
||
|
Response 1: Thank you for pointing this out. I/We agree with this comment. All the isolates included in the study were carbapenem-resistant; we made a statement initially in the abstract section and then in the methodology sections (lines 37, 145). Not all the isolates from the hospital over the study period were carbapenem-resistant. However, isolates sensitive to carbapenems were excluded from the study. Also, only the first isolate was included in case of reinfection and patients with records available of around 90 days after a positive culture were included. In this study we did not report the frequency of the CRPA isolation. However previous studies shown that from 2018 to 2020, the frequency of isolating P. aeruginosa in the hospital was about 5.64%, increasing to 10.08% from 2020 to 2022. During the period 2018-2020, 35.03% of the isolates were CRPA, which rose to 78.76% from 2020-2022 but dropped to around 46% by the end of 2022. References: · Martínez-Ponce C.A. 2023. Comparación de microbiologia y susceptibilidad antimicrobiana durate periodo de 2018-2020 y periodo 2020-2022 dentro del Centenairo Hospital Miguel Hidalgo. Tesis de Especialidades. Universidad Autónoma de Aguascalientes. Library repository UAA. URI: http://hdl.handle.net/11317/2914
|
||
|
|
||
|
|
||
|
2. Comment 2: All of the isolates were from the same hospital. Were all of the isolates from different patients? As they were nosocomial infections, there is the possibility of a common source of infection which could cause identical bacteria to be isolated from different patients. The inadvertant inclusion of multiple samples of the same isolate(s) would affect the statistical analysis. Did the authors carry out any typing to show that all isolates were independent and unrelated? If not, the possibility that duplicates (or more) of the same isolate might have been included and the impact that would have on results needs to be discussed.
Response 2: We thank the reviewer for this comment. All isolates were obtained from the same hospital. This study was observational and retrospective, conducted at a single healthcare center. Each isolate was sourced from a different patient. To ensure that no duplicate strains were included, only one isolate per patient was considered in the study. We have now clarified that point in the abstract and the methodology. The reviewer noted that a common source of infection might lead to the isolation of an identical bacteria clone from different patients. We cannot confirm that all the strains were unique, as one limitation of the study is that we did not perform any typing. However, the study revealed different genetic characteristics in the isolates. We included only one isolation per patient, and the strains were obtained from various specimen sources and hospital areas over an extended isolation period of nearly four years. Therefore, we suggest that the bacterial isolates are different among patients. This observation was mentioned in the discussion section (lines 317-322) and methodology section (lines 145).
3. Comment 3: Antibiotic susceptibility testing – what were the resistance/ intermediate MIC threshold for antibiotics other than carbapenems?
Response 3: We thank the reviewer for this comment. All MIC values are included in the supplementary data, specifically in Supplementary Table S2, which contains the breakpoint values of the antimicrobial agents utilized in this study. We followed the CLSI guidelines M100 from 2020.
4. Comment 4: Table 1. What is “reference”? “Odd ratio” should presumably be “Odds ratio”, as the full term is used in the table heading it is not necessary to define “OR” in the footnote.
Response 4: We removed the OR from the table heading and added the “s” missing in the Odds ratio.
5. Comment 5: Line 212. What is “Non-blaOXA23”?
Response 5: We clarified this statement. Now it says: “None of the isolates tested carried blaOXA-23”.
6. Comment 6: Table 3. The p-value describes (presumably) whether there is a significant difference between ciprofloxacin-resistant and -sensitive samples. As only carbapenem-resistant isolates were included in the study, it is meaningless to assess the difference between cipro-resistant and -sensitive isolates and the lines for imipenem, meropenem and doripenem can be deleted.
Response 6: We agree with the reviewer. The lines for imipenem, meropenem and doripenem in Table 3 were deleted.
7. Comment 7: Fluoroquinolone resistance. Table 3 indicates that a lot of the ciprofloxacin-resistant isolates did not have any PMQR genes. Presumably these isolates are resistant due to mutations in gyrA or other genes associated with resistance. In addition, oqxA is present in ciprofloxacin-sensitive strains. The authors should discuss whether the isolates that they studied also contain mutations in gyrA or other genes, and the extent to which the PMQR genes contribute to the ciprofloxacin-resistance phenotype.
Response 7: We appreciate the reviewer’s comment. As noted, the isolates that exhibited ciprofloxacin resistance but did not carry any plasmid-mediated quinolone resistance (PMQR) genes may be due to mutations in the gyrA and/or parC genes. In this study, we observed minimum inhibitory concentration (MIC) values of ≥ 4 µg/ml, suggesting that the strains likely possess a single mutation in either gyrA or parC. This conclusion is based on the understanding that low-level resistance is typically associated with a single mutation in one gene, while high-level resistance usually involves mutations in both gyrA and parC simultaneously. Future studies will focus on examining mutations in gyrA, parC, gyrB, and parE to better understand the implications of these mutations on MIC values and to complement this study. We pointed out in the discussion section (lines 371-376). In this study, we examined the oqxA gene in both ciprofloxacin-resistant and ciprofloxacin-sensitive isolates. Our findings suggest that oqxA contributes to a low level of resistance in these isolates, with minimum inhibitory concentration (MIC) values not exceeding 4 µg/ml. Interestingly, all the strains tested were resistant to tigecycline, which may be linked to the presence of oqx genes, as previous research has shown that these genes can confer low to intermediate resistance to tigecycline. We pointed out in the discussion section (lines 380-395).
8. Comment 8: Lines 334-352, presence of PMQR and b lactamase genes in the same isolates. Do the authors think that the PMQR genes and b lactamase genes are present on the same genetic elements in the isolates in their study?
Response 8: It was not the aim of the present study to determine whether the carbapenemase-encoding genes and the PMQR genes were in the same genetic element. However, some studies indicate that IncF plasmids carry a diverse array of resistance genes for major classes of antimicrobials, particularly Extended-Spectrum Beta-Lactamase (ESBL) genes like those of the CTX-M type, Plasmid-mediated Quinolone Resistance (PMQR) genes, and genes encoding aminoglycoside-modifying enzymes such as aac-(6’)-Ib-cr in Escherichia coli (Rozwandowicz et al., 2018). Notably, the IncF plasmid carrying the aac-(6’)-Ib-cr gene has been identified in P.s aeruginosa (Ogbolu et al., 2013). Additionally, other research has documented the co-existence of the qnrB gene with other resistance genes, such as blaCTX-M-14 or blaCTX-M-15, on the same plasmid derived from animal sources. An example is the IncFII plasmid that contains a multidrug resistance region with blaCTX-M-15, blaTEM-1B, blaOXA-1, aac(6′)-Ib-cr, and qnrB2 (Strahilevitz et al., 2009; Pomba et al., 2009). Thus, it could be suggesting the possibility that the presence of the beta-lactamase genes and PMQR are in the same plasmid. In other hand, bacteria can acquire new factors that facilitate the development of multidrug resistance since development of antimicrobial resistance in this bacterium may be attributed to the excessive and inappropriate use of antibiotics, thus, under selective pressure, can induce response facilitates bacterial survival and develops and acquire antibiotic resistance. Moreover, carbapenemase production encoded by genes located on mobile genetic elements is typically accompanied by genetic encoding resistance to other drug classes, but not necessarily located on the same mobile element. We point it out in the discussion section (lines 422-432).
9. Comment 9: Why do their results indicate “dissemination of PMQR genes on carbapenem-resistant strains”, why not dissemination of b lactamase genes on ciprofloxacin-resistant strains? This paragraph is something of a catalogue, documenting other studies – what is the take-home message?
Response 9: In this study our inclusion criteria were that all the strain tested presented carbapenem resistance (imipenem, meropenem or doripenem). Thus, in the light of that point, PMQR genes are present in carbapenem-resistant strains.
10. Comment 10: In connection with that point, the study would be greatly strengthened by analysis of PMQR genes in carbapenem-sensitive isolates from the same hospital – is the frequency of PMQR genes lower in isolates that are carbapenem-sensitive but ciprofloxacin-resistant?
Response 10: We appreciate the reviewer's comment and fully agree with their perspective. We believe that our study may improve by analyzing PMQR genes in carbapenem-sensitive isolates from the same hospital. Numerous studies have shown that the prevalence of plasmid-mediated quinolone resistance (PMQR) genes is often higher among phenotypically fluoroquinolone-resistant isolates. For instance, a study by Kotb et al. (2019) examined the bacteria Escherichia coli, Klebsiella pneumoniae, and Citrobacter. In this study, all isolates with high and intermediate resistance phenotypes carried one or more PMQR genes, while those isolates exhibiting a fluoroquinolone-susceptible phenotype harbored none of the tested PMQR genes (including qnrA, qnrB, qnrC, qnrS, and qepA). Another study by Abdelrahim et al., 2024, found a positive association between fluoroquinolone resistant isolates of P. aeruginosa and imipenem resistance and cefepime resistance. Indeed, they found a significant difference between ciprofloxacin resistant vs ciprofloxacin sensitive isolates, and the presence of ESBL genes, being higher the frequency of ESBL genes in the ciprofloxacin resistant group (p=0.003), and imipinemem resistant isolates (p = 0.002). Moreover, fluroquinolone resistant isolates showed significantly higher carriage of the blaCTX-M gene than sensitive ones (p = 0.003), and a high association of aac(6’)lb cr gene with ESBL genes. Szabo et al. (2018) found that PMQR (plasmid-mediated quinolone resistance) genes were present in isolates with susceptibility or low-level resistance to ciprofloxacin, with minimum inhibitory concentrations (MIC) ranging from 0.06 to 1 mg/L. Piekarska et al. (2015) indicated that the combination of PMQR genes and mutations in the quinolone resistance-determining regions (QRDRs) of the gyrA and parC genes contributes to high resistance to fluoroquinolones. In a study by Venkataramana et al. (2022), it was found that 78.8% of ciprofloxacin-resistant P. aeruginosa isolates carried PMQR genes, with the acc(6’)-lb-cr gene being the most prevalent with 36.4%. The MICs of ciprofloxacin for these isolates ranged from 4 to ≥ 256 µg/mL. Additionally, another study focusing on Klebsiella pneumoniae conducted by Venkataramana et al. (2020) reported high frequencies of several PMQR genes: qnrB (12%), qnrS (4.5%), aac(6’)-lb-cr (89%), and oqxAB (6.3%) among strains exhibiting high-level of ciprofloxacin resistance, with MICs ranging from 4 µg/mL to ≥ 256 µg/mL. Moreover, 85% of these strains harbored mutations in gyrA and gyrB, while 80% had mutations in parC. These findings suggest a higher frequency of PMQR genes in strains with elevated MIC levels for ciprofloxacin resistance, as well as the presence of multiple mutations in the QRDR regions Altogether, these studies indicate that carbapenem-resistant isolates also exhibit resistance to fluoroquinolones. As a result, it is likely that we will find a higher prevalence of plasmid-mediated quinolone resistance (PMQR) genes and mutations in the quinolone determinant region (QRDR) within these strains. In contrast, carbapenemase-sensitive strains showed only a low level of fluoroquinolone resistance, suggesting that PMQR may be less common in these cases. Further research is needed to explore this aspect in the future. References: · Piekarska K, Wołkowicz T, Zacharczuk K, et al. Co-existence of plasmid mediated quinolone resistance determinants and mutations in gyrA and parC among fluoroquinolone-resistant clinical Enterobacteriaceae isolated in a tertiary hospital in Warsaw, Poland. Int J Antimicrob Agents. 2015;45(3):238–43 · Szabó O, Gulyás D, Szabó N, et al. Plasmid-mediated quinolone resistance determinants in Enterobacteriaceae from urine clinical samples. Acta Microbiol Immunol Hung. 2018;65:231–11. · Kotb, D.N., Mahdy, W.K., Mahmoud, M.S. et al. Impact of co-existence of PMQR genes and QRDR mutations on fluoroquinolones resistance in Enterobacteriaceae strains isolated from community and hospital acquired UTIs. BMC Infect Dis 19, 979 (2019). https://doi.org/10.1186/s12879-019-4606-y · Venkataramana GP, Lalitha AKV, Mariappan S, Sekar U. Plasmid-Mediated Fluoroquinolone Resistance in Pseudomonas aeruginosa and Acinetobacter baumannii. J Lab Physicians. · Geetha PV, Aishwarya KVL, Mariappan S, Sekar U. Fluoroquinolone Resistance in Clinical Isolates of Klebsiella Pneumonia e. J Lab Physicians. 2020;12(2):121-125. doi:10.1055/s-0040-17164782022 Feb 9;14(3):271-277. doi: 10.1055/s-0042-1742636. PMID: 36119417; PMCID: PMC9473940.
11. Comment 11: Why would understanding the mechanism of carbapenem-resistance be “invaluable to take measures and prevent the spread of emerging resistance mechanisms in hospital environments” (line 357)? – please explain.
Response 11: We delated this line in order to avoid misunderstandings. We think that the manuscript it contains valuable information, as the mechanisms of carbapenem resistance can significantly influence the best treatment option and the time it takes to initiate therapy. Confirming and distinguishing among the various classes of carbapenemases is critical to start an appropriate early treatment, as not all beta-lactamases share the same treatment options. |
||
|
|
||
|
|
||
|
5. Additional clarifications |
||
|
Other modifications implemented in based on the reviewers’ comments: 1. We compared the national and global antimicrobial resistance data in the discussion section and in our study (lines 291-304). 2. We have added an illustration at the end of the manuscript in order to summarize the main PMQR determinants and carbapenemase-encoding genes found in our study (line 447, Figure 1). We have also added a graphical abstract of the study. 3. We reviewed the typing errors and polished the entire manuscript. 4. We cross-checked all the abbreviations in the manuscript and defined them initially with their respective full name, followed by an abbreviation. 5. We emphasized the clinical significance of this co-occurrence in the introduction (lines 83 to 98). 6. We added this issue at the end of the discussion (line 438-445) |
||
Reviewer 2 Report
Comments and Suggestions for Authors
Tapia-Cornejo et al. retrospectively investigated the prevalence of PMQR and carbapenemase-encoding genes in fifty-two clinical P. aeruginosa clinical isolates from nosocomial patients in Centenario Hospital Miguel Hidalgo, Aguascalientes State, Mexico. Although the study is limited to a single center, it is envisaged to guide the rational prescription of antibiotics in a regional context.
Minor comment:
I suggest the author indicate in the abstract the observational study design and the fact that it is a single-center study.
Author Response
For research article
|
Response to Reviewer 2 Comments
|
||
|
1. Summary |
|
|
|
Thank you very much for taking the time to review this manuscript. Please find the detailed responses below and the corresponding revisions/corrections highlighted/in track changes in the re-submitted files. We appreciate all your valuable comments and suggestions, which have helped us to improve the quality of the manuscript. We considered carefully each of the comments and tried our best to answer and solve them. We have incorporated into the new version of the manuscript most of the reviewers’ suggestions, and the changes are highlighted within the manuscript. We sincerely hope that you will find our revised manuscript adequate for publication.
We break down the answers given to each reviewer below.
|
||
|
2. Questions for General Evaluation |
Reviewer’s Evaluation |
Response and Revisions |
|
Does the introduction provide sufficient background and include all relevant references? |
Yes/Can be improved/Must be improved/Not applicable |
We response in the point-by-point response letter. |
|
Are all the cited references relevant to the research? |
Yes/Can be improved/Must be improved/Not applicable |
|
|
Is the research design appropriate? |
Yes/Can be improved/Must be improved/Not applicable |
|
|
Are the methods adequately described? |
Yes/Can be improved/Must be improved/Not applicable |
|
|
Are the results clearly presented? |
Yes/Can be improved/Must be improved/Not applicable |
|
|
Are the conclusions supported by the results? |
Yes/Can be improved/Must be improved/Not applicable |
|
|
3. Point-by-point response to Comments and Suggestions for Authors
|
||
|
1. Comment 1: I suggest the author indicate in the abstract the observational study design and the fact that it is a single-center study.
Response 1: We thank the reviewer for this comment. We are now pointing out that the study's design was observational, retrospective and from a single health-care center in the abstract and methodology sections (line 34 – 35).
|
||
|
|
||
|
5. Additional clarifications |
||
|
All modifications implemented in based on the reviewers’ comments: 1. We compared the national and global antimicrobial resistance data in the discussion section and in our study (lines 291-304). 2. We have added an illustration at the end of the manuscript in order to summarize the main PMQR determinants and carbapenemase-encoding genes found in our study (line 447, Figure 1). We have also added a graphical abstract of the study. 3. We reviewed the typing errors and polished the entire manuscript. 4. We cross-checked all the abbreviations in the manuscript and defined them initially with their respective full name, followed by an abbreviation. 5. We emphasized the clinical significance of this co-occurrence in the introduction (lines 83 to 98). 6. We added this issue at the end of the discussion (line 438-445). 7. This section was not included in the manuscript; however we pointed out this information for all the reviewers: All the isolates included in the study were carbapenem-resistant; we made a statement initially in the abstract section and then in the methodology sections (lines 37, 145). Not all the isolates from the hospital over the study period were carbapenem-resistant. However, isolates sensitive to carbapenems were excluded from the study. Also, only the first isolate was included in case of reinfection and patients with records available of around 90 days after a positive culture were included. In this study we did not report the frequency of the CRPA isolation. However previous studies shown that from 2018 to 2020, the frequency of isolating P. aeruginosa in the hospital was about 5.64%, increasing to 10.08% from 2020 to 2022. During the period 2018-2020, 35.03% of the isolates were CRPA, which rose to 78.76% drom 2020-2022 but dropped to around 46% by the end of 2022. 8. All isolates were obtained from the same hospital. This study was observational and retrospective, conducted at a single healthcare center. Each isolate was sourced from a different patient. To ensure that no duplicate strains were included, only one isolate per patient was considered in the study. We have now clarified that point in the abstract and the methodology. 9. The reviewer noted that a common source of infection might lead to the isolation of an identical bacteria clone from different patients. We cannot confirm that all the strains were unique, as one limitation of the study is that we did not perform any typing. However, the study revealed different genetic characteristics in the isolates. We included only one isolation per patient, and the strains were obtained from various specimen sources and hospital areas over an extended isolation period of nearly four years. Therefore, we suggest that the bacterial isolates are different among patients. This observation was mentioned in the discussion section (lines 317-322) and methodology section (lines 145). 10. All MIC values are included in the supplementary data, specifically in Supplementary Table S2, which contains the breakpoint values of the antimicrobial agents utilized in this study. We followed the CLSI guidelines M100 from 2020. 11. We removed the OR from the table heading and added the “s” missing in the Odds ratio. 12. We clarified this statement. Now it says: “None of the isolates tested carried blaOXA-23”. 13. In the table 3, The lines for imipenem, meropenem and doripenem in Table 3 were deleted. 14. As noted, the isolates that exhibited ciprofloxacin resistance but did not carry any plasmid-mediated quinolone resistance (PMQR) genes may be due to mutations in the gyrA and/or parC genes. In this study, we observed minimum inhibitory concentration (MIC) values of ≥ 4 µg/ml, suggesting that the strains likely possess a single mutation in either gyrA or parC. This conclusion is based on the understanding that low-level resistance is typically associated with a single mutation in one gene, while high-level resistance usually involves mutations in both gyrA and parC simultaneously. Future studies will focus on examining mutations in gyrA, parC, gyrB, and parE to better understand the implications of these mutations on MIC values and to complement this study. We pointed out in the discussion section (lines 371-376). In this study, we examined the oqxA gene in both ciprofloxacin-resistant and ciprofloxacin-sensitive isolates. Our findings suggest that oqxA contributes to a low level of resistance in these isolates, with minimum inhibitory concentration (MIC) values not exceeding 4 µg/ml. Interestingly, all the strains tested were resistant to tigecycline, which may be linked to the presence of oqx genes, as previous research has shown that these genes can confer low to intermediate resistance to tigecycline. We pointed out in the discussion section (lines 380-395). 15. It was not the aim of the present study to determine whether the carbapenemase-encoding genes and the PMQR genes were in the same genetic element. However, some studies indicate that IncF plasmids carry a diverse array of resistance genes for major classes of antimicrobials, particularly Extended-Spectrum Beta-Lactamase (ESBL) genes like those of the CTX-M type, Plasmid-mediated Quinolone Resistance (PMQR) genes, and genes encoding aminoglycoside-modifying enzymes such as aac-(6’)-Ib-cr in Escherichia coli (Rozwandowicz et al., 2018). Notably, the IncF plasmid carrying the aac-(6’)-Ib-cr gene has been identified in Pseudomonas aeruginosa (Ogbolu et al., 2013). Additionally, other research has documented the co-existence of the qnrB gene with other resistance genes, such as blaCTX-M-14 or blaCTX-M-15, on the same plasmid derived from animal sources. An example is the IncFII plasmid that contains a multidrug resistance region with blaCTX-M-15, blaTEM-1B, blaOXA-1, aac(6′)-Ib-cr, and qnrB2 (Strahilevitz et al., 2009; Pomba et al., 2009). Thus, it could be suggesting the possibility that the presence of the beta-lactamase genes and PMQR are in the same plasmid. In other hand, bacteria can acquire new factors that facilitate the development of multidrug resistance since development of antimicrobial resistance in this bacterium may be attributed to the excessive and inappropriate use of antibiotics, thus, under selective pressure, can induce response facilitates bacterial survival and develops and acquire antibiotic resistance. Moreover, carbapenemase production encoded by genes located on mobile genetic elements is typically accompanied by genetic encoding resistance to other drug classes, but not necessarily located on the same mobile element. We point it out in the discussion section (lines 422-432). 16. In this study our inclusion criteria were that all the strain tested presented carbapenem resistance (imipenem, meropenem or doripenem). Thus, in the light of that point, PMQR genes are present in carbapenem-resistant strains. 17. We believe that our study may improve by analyzing PMQR genes in carbapenem-sensitive isolates from the same hospital. 18. Numerous studies have shown that the prevalence of plasmid-mediated quinolone resistance (PMQR) genes is often higher among phenotypically fluoroquinolone-resistant isolates. For instance, a study by Kotb et al. (2019) examined the bacteria Escherichia coli, Klebsiella pneumoniae, and Citrobacter. In this study, all isolates with high and intermediate resistance phenotypes carried one or more PMQR genes, while those isolates exhibiting a fluoroquinolone-susceptible phenotype harbored none of the tested PMQR genes (including qnrA, qnrB, qnrC, qnrS, and qepA). Another study by Abdelrahim et al., 2024, found a positive association between fluoroquinolone resistant isolates of P. aeruginosa and imipenem resistance and cefepime resistance. Indeed, they found a significant difference between ciprofloxacin resistant vs ciprofloxacin sensitive isolates, and the presence of ESBL genes, being higher the frequency of ESBL genes in the ciprofloxacin resistant group (p=0.003), and imipinemem resistant isolates (p = 0.002). Moreover, fluroquinolone resistant isolates showed significantly higher carriage of the blaCTX-M gene than sensitive ones (p = 0.003), and a high association of aac(6’)lb cr gene with ESBL genes. Szabo et al. (2018) found that PMQR (plasmid-mediated quinolone resistance) genes were present in isolates with susceptibility or low-level resistance to ciprofloxacin, with minimum inhibitory concentrations (MIC) ranging from 0.06 to 1 mg/L. Piekarska et al. (2015) indicated that the combination of PMQR genes and mutations in the quinolone resistance-determining regions (QRDRs) of the gyrA and parC genes contributes to high resistance to fluoroquinolones. In a study by Venkataramana et al. (2022), it was found that 78.8% of ciprofloxacin-resistant P. aeruginosa isolates carried PMQR genes, with the acc(6’)-lb-cr gene being the most prevalent with 36.4%. The MICs of ciprofloxacin for these isolates ranged from 4 to ≥ 256 µg/mL. Additionally, another study focusing on Klebsiella pneumoniae conducted by Venkataramana et al. (2020) reported high frequencies of several PMQR genes: qnrB (12%), qnrS (4.5%), aac(6’)-lb-cr (89%), and oqxAB (6.3%) among strains exhibiting high-level of ciprofloxacin resistance, with MICs ranging from 4 µg/mL to ≥ 256 µg/mL. Moreover, 85% of these strains harbored mutations in gyrA and gyrB, while 80% had mutations in parC. These findings suggest a higher frequency of PMQR genes in strains with elevated MIC levels for ciprofloxacin resistance, as well as the presence of multiple mutations in the QRDR regions. Altogether, these studies indicate that carbapenem-resistant isolates also exhibit resistance to fluoroquinolones. As a result, it is likely that we will find a higher prevalence of plasmid-mediated quinolone resistance (PMQR) genes and mutations in the quinolone determinant region (QRDR) within these strains. In contrast, carbapenemase-sensitive strains showed only a low level of fluoroquinolone resistance, suggesting that PMQR may be less common in these cases. Further research is needed to explore this aspect in the future. 19. We think that the manuscript contains valuable information, as the mechanisms of carbapenem resistance can significantly influence the best treatment option and the time it takes to initiate therapy. Confirming and distinguishing among the various classes of carbapenemases is critical to start an appropriate early treatment, as not all beta-lactamases share the same treatment options.
|
||
Reviewer 3 Report
Comments and Suggestions for Authors
This study “Occurrence of Plasmid-Mediated Quinolone Resistance and carbapenemase-encoding genes in Pseudomonas aeruginosa isolates from nosocomial patients in Aguascalientes, Mexico” is a good read. This study sought to examine the co-occurrence of Plasmid-Mediated Quinolone Resistance (PMQR) determinants and carbapenemase-encoding genes in P. aeruginosa strains obtained from nosocomial patients in Aguascalientes, Mexico. Fifty-two isolates were obtained, and antibiotic susceptibility was assessed with the Vitek-2 technique, accompanied by PCR screening for resistance genes. The isolates exhibited complete resistance to carbapenems, tigecycline, and ceftriaxone. Of the strains, 34.6% possessed qnr genes, 46.2% included the oqxA gene, and 25% exhibited the aac-(6')-lb gene. The predominant carbapenemase genes identified were blaOXA-51 (42.3%), blaOXA-1 (15.4%), and blaVIM (15.4%). The co-occurrence of resistance genes was noted, with blaOXA-51 co-existing with oqxA in 21.1% of isolates. The research underscores the necessity for comprehensive monitoring of resistance genes and prudent application of last-resort antibiotics to mitigate the proliferation of multidrug-resistant bacteria.
Altogether this is an important and timely article, this reviewer has certain suggestions that would help produce a more comprehensive overview of the topic:
Comments:
1, The study provides significant resistance data; nevertheless, a more comprehensive statistical analysis or comparisons with other places or prior studies would enhance the discussion. How do these results align with national or global trends?
2, At least one additional Figure (illustration) may be provided as to highlight the summary or prospect of this study.
3, The English of manuscript can be polished (minor) and there are few typological errors.
4, The author should cross-check all abbreviations in the manuscript. Initially, define in full name followed by abbreviation.
5, The study aims to investigate the co-occurrence of PMQR determinants and carbapenemase-encoding genes in Pseudomonas aeruginosa isolates. Emphasizing the clinical significance of this co-occurrence might enhance the introduction. What makes this particular co-occurrence troubling about treatment alternatives?
6, Authors should add a paragraph to discuss limitations of this study.
Author Response
For research article
|
Response to Reviewer 3 Comments
|
||
|
1. Summary |
|
|
|
Thank you very much for taking the time to review this manuscript. Please find the detailed responses below and the corresponding revisions/corrections highlighted/in track changes in the re-submitted files. We appreciate all your valuable comments and suggestions, which have helped us to improve the quality of the manuscript. We considered carefully each of the comments and tried our best to answer and solve them. We have incorporated into the new version of the manuscript most of the reviewers’ suggestions, and the changes are highlighted within the manuscript. We sincerely hope that you will find our revised manuscript adequate for publication.
We break down the answers given to each reviewer below.
|
||
|
2. Questions for General Evaluation |
Reviewer’s Evaluation |
Response and Revisions |
|
Does the introduction provide sufficient background and include all relevant references? |
Yes/Can be improved/Must be improved/Not applicable |
We response in the point-by-point response letter. |
|
Are all the cited references relevant to the research? |
Yes/Can be improved/Must be improved/Not applicable |
|
|
Is the research design appropriate? |
Yes/Can be improved/Must be improved/Not applicable |
|
|
Are the methods adequately described? |
Yes/Can be improved/Must be improved/Not applicable |
|
|
Are the results clearly presented? |
Yes/Can be improved/Must be improved/Not applicable |
|
|
Are the conclusions supported by the results? |
Yes/Can be improved/Must be improved/Not applicable |
|
|
3. Point-by-point response to Comments and Suggestions for Authors
|
||
|
1. Comment 1: The study provides significant resistance data; nevertheless, a more comprehensive statistical analysis or comparisons with other places or prior studies would enhance the discussion. How do these results align with national or global trends?
Response: We compared the national and global antimicrobial resistance data in the discussion section and in our study (lines 291-304).
2. Comment 2: At least one additional Figure (illustration) may be provided as to highlight the summary or prospect of this study.
Response: We have added an illustration at the end of the manuscript in order to summarize the main PMQR determinants and carbapenemase-encoding genes found in our study (line 447, Figure 1). We have also added a graphical abstract of the study.
3. Comment 3: The English of manuscript can be polished (minor) and there are few typological errors.
Response: We reviewed the typing errors and polished the entire manuscript.
4. Comment 4: The author should cross-check all abbreviations in the manuscript. Initially, define in full name followed by abbreviation.
Response: We cross-checked all the abbreviations in the manuscript and defined them initially with their respective full name, followed by an abbreviation.
5. Comment 5: The study aims to investigate the co-occurrence of PMQR determinants and carbapenemase-encoding genes in Pseudomonas aeruginosa isolates. Emphasizing the clinical significance of this co-occurrence might enhance the introduction. What makes this particular co-occurrence troubling about treatment alternatives?
Response: We thank the reviewer for this comment. We emphasized the clinical significance of this co-occurrence in the introduction (lines 83 to 98).
6. Comment 6: Authors should add a paragraph to discuss limitations of this study.
Response: We thank the reviewer for this comment. We added this issue at the end of the discussion (line 438-445).
|
||
|
|
||
|
5. Additional clarifications |
||
|
All modifications implemented in based on the reviewers’ comments: 1. We compared the national and global antimicrobial resistance data in the discussion section and in our study (lines 291-304). 2. We have added an illustration at the end of the manuscript in order to summarize the main PMQR determinants and carbapenemase-encoding genes found in our study (line 447, Figure 1). We have also added a graphical abstract of the study. 3. We reviewed the typing errors and polished the entire manuscript. 4. We cross-checked all the abbreviations in the manuscript and defined them initially with their respective full name, followed by an abbreviation. 5. We emphasized the clinical significance of this co-occurrence in the introduction (lines 83 to 98). 6. We added this issue at the end of the discussion (line 438-445). 7. This section was not included in the manuscript; however we pointed out this information for all the reviewers: All the isolates included in the study were carbapenem-resistant; we made a statement initially in the abstract section and then in the methodology sections (lines 37, 145). Not all the isolates from the hospital over the study period were carbapenem-resistant. However, isolates sensitive to carbapenems were excluded from the study. Also, only the first isolate was included in case of reinfection and patients with records available of around 90 days after a positive culture were included. In this study we did not report the frequency of the CRPA isolation. However previous studies shown that from 2018 to 2020, the frequency of isolating P. aeruginosa in the hospital was about 5.64%, increasing to 10.08% from 2020 to 2022. During the period 2018-2020, 35.03% of the isolates were CRPA, which rose to 78.76% drom 2020-2022 but dropped to around 46% by the end of 2022. 8. All isolates were obtained from the same hospital. This study was observational and retrospective, conducted at a single healthcare center. Each isolate was sourced from a different patient. To ensure that no duplicate strains were included, only one isolate per patient was considered in the study. We have now clarified that point in the abstract and the methodology. 9. The reviewer noted that a common source of infection might lead to the isolation of an identical bacteria clone from different patients. We cannot confirm that all the strains were unique, as one limitation of the study is that we did not perform any typing. However, the study revealed different genetic characteristics in the isolates. We included only one isolation per patient, and the strains were obtained from various specimen sources and hospital areas over an extended isolation period of nearly four years. Therefore, we suggest that the bacterial isolates are different among patients. This observation was mentioned in the discussion section (lines 317-322) and methodology section (lines 145). 10. All MIC values are included in the supplementary data, specifically in Supplementary Table S2, which contains the breakpoint values of the antimicrobial agents utilized in this study. We followed the CLSI guidelines M100 from 2020. 11. We removed the OR from the table heading and added the “s” missing in the Odds ratio. 12. We clarified this statement. Now it says: “None of the isolates tested carried blaOXA-23”. 13. In the table 3, The lines for imipenem, meropenem and doripenem in Table 3 were deleted. 14. As noted, the isolates that exhibited ciprofloxacin resistance but did not carry any plasmid-mediated quinolone resistance (PMQR) genes may be due to mutations in the gyrA and/or parC genes. In this study, we observed minimum inhibitory concentration (MIC) values of ≥ 4 µg/ml, suggesting that the strains likely possess a single mutation in either gyrA or parC. This conclusion is based on the understanding that low-level resistance is typically associated with a single mutation in one gene, while high-level resistance usually involves mutations in both gyrA and parC simultaneously. Future studies will focus on examining mutations in gyrA, parC, gyrB, and parE to better understand the implications of these mutations on MIC values and to complement this study. We pointed out in the discussion section (lines 371-376). In this study, we examined the oqxA gene in both ciprofloxacin-resistant and ciprofloxacin-sensitive isolates. Our findings suggest that oqxA contributes to a low level of resistance in these isolates, with minimum inhibitory concentration (MIC) values not exceeding 4 µg/ml. Interestingly, all the strains tested were resistant to tigecycline, which may be linked to the presence of oqx genes, as previous research has shown that these genes can confer low to intermediate resistance to tigecycline. We pointed out in the discussion section (lines 380-395). 15. It was not the aim of the present study to determine whether the carbapenemase-encoding genes and the PMQR genes were in the same genetic element. However, some studies indicate that IncF plasmids carry a diverse array of resistance genes for major classes of antimicrobials, particularly Extended-Spectrum Beta-Lactamase (ESBL) genes like those of the CTX-M type, Plasmid-mediated Quinolone Resistance (PMQR) genes, and genes encoding aminoglycoside-modifying enzymes such as aac-(6’)-Ib-cr in Escherichia coli (Rozwandowicz et al., 2018). Notably, the IncF plasmid carrying the aac-(6’)-Ib-cr gene has been identified in Pseudomonas aeruginosa (Ogbolu et al., 2013). Additionally, other research has documented the co-existence of the qnrB gene with other resistance genes, such as blaCTX-M-14 or blaCTX-M-15, on the same plasmid derived from animal sources. An example is the IncFII plasmid that contains a multidrug resistance region with blaCTX-M-15, blaTEM-1B, blaOXA-1, aac(6′)-Ib-cr, and qnrB2 (Strahilevitz et al., 2009; Pomba et al., 2009). Thus, it could be suggesting the possibility that the presence of the beta-lactamase genes and PMQR are in the same plasmid. In other hand, bacteria can acquire new factors that facilitate the development of multidrug resistance since development of antimicrobial resistance in this bacterium may be attributed to the excessive and inappropriate use of antibiotics, thus, under selective pressure, can induce response facilitates bacterial survival and develops and acquire antibiotic resistance. Moreover, carbapenemase production encoded by genes located on mobile genetic elements is typically accompanied by genetic encoding resistance to other drug classes, but not necessarily located on the same mobile element. We point it out in the discussion section (lines 422-432). 16. In this study our inclusion criteria were that all the strain tested presented carbapenem resistance (imipenem, meropenem or doripenem). Thus, in the light of that point, PMQR genes are present in carbapenem-resistant strains. 17. We believe that our study may improve by analyzing PMQR genes in carbapenem-sensitive isolates from the same hospital. 18. Numerous studies have shown that the prevalence of plasmid-mediated quinolone resistance (PMQR) genes is often higher among phenotypically fluoroquinolone-resistant isolates. For instance, a study by Kotb et al. (2019) examined the bacteria Escherichia coli, Klebsiella pneumoniae, and Citrobacter. In this study, all isolates with high and intermediate resistance phenotypes carried one or more PMQR genes, while those isolates exhibiting a fluoroquinolone-susceptible phenotype harbored none of the tested PMQR genes (including qnrA, qnrB, qnrC, qnrS, and qepA). Another study by Abdelrahim et al., 2024, found a positive association between fluoroquinolone resistant isolates of P. aeruginosa and imipenem resistance and cefepime resistance. Indeed, they found a significant difference between ciprofloxacin resistant vs ciprofloxacin sensitive isolates, and the presence of ESBL genes, being higher the frequency of ESBL genes in the ciprofloxacin resistant group (p=0.003), and imipinemem resistant isolates (p = 0.002). Moreover, fluroquinolone resistant isolates showed significantly higher carriage of the blaCTX-M gene than sensitive ones (p = 0.003), and a high association of aac(6’)lb cr gene with ESBL genes. Szabo et al. (2018) found that PMQR (plasmid-mediated quinolone resistance) genes were present in isolates with susceptibility or low-level resistance to ciprofloxacin, with minimum inhibitory concentrations (MIC) ranging from 0.06 to 1 mg/L. Piekarska et al. (2015) indicated that the combination of PMQR genes and mutations in the quinolone resistance-determining regions (QRDRs) of the gyrA and parC genes contributes to high resistance to fluoroquinolones. In a study by Venkataramana et al. (2022), it was found that 78.8% of ciprofloxacin-resistant P. aeruginosa isolates carried PMQR genes, with the acc(6’)-lb-cr gene being the most prevalent with 36.4%. The MICs of ciprofloxacin for these isolates ranged from 4 to ≥ 256 µg/mL. Additionally, another study focusing on Klebsiella pneumoniae conducted by Venkataramana et al. (2020) reported high frequencies of several PMQR genes: qnrB (12%), qnrS (4.5%), aac(6’)-lb-cr (89%), and oqxAB (6.3%) among strains exhibiting high-level of ciprofloxacin resistance, with MICs ranging from 4 µg/mL to ≥ 256 µg/mL. Moreover, 85% of these strains harbored mutations in gyrA and gyrB, while 80% had mutations in parC. These findings suggest a higher frequency of PMQR genes in strains with elevated MIC levels for ciprofloxacin resistance, as well as the presence of multiple mutations in the QRDR regions. Altogether, these studies indicate that carbapenem-resistant isolates also exhibit resistance to fluoroquinolones. As a result, it is likely that we will find a higher prevalence of plasmid-mediated quinolone resistance (PMQR) genes and mutations in the quinolone determinant region (QRDR) within these strains. In contrast, carbapenemase-sensitive strains showed only a low level of fluoroquinolone resistance, suggesting that PMQR may be less common in these cases. Further research is needed to explore this aspect in the future. 19. We think that the manuscript contains valuable information, as the mechanisms of carbapenem resistance can significantly influence the best treatment option and the time it takes to initiate therapy. Confirming and distinguishing among the various classes of carbapenemases is critical to start an appropriate early treatment, as not all beta-lactamases share the same treatment options.
|
||
Round 2
Reviewer 1 Report
Comments and Suggestions for Authors
I have no additional comments.